# One For All & All For One: Bypassing Hyperparameter Tuning with Model Averaging For Cross-Lingual Transfer

**Fabian David Schmidt[1], Ivan Vulić[2], Goran Glavaš[1]**

[1] Center For Artificial Intelligence and Data Science, University of Würzburg, Germany
[2] Language Technology Lab, University of Cambridge, UK
{fabian.schmidt, goran.glavas}@uni-wuerzburg.de
iv250@cam.ac.uk

## Abstract

Multilingual language models enable zero-shot cross-lingual transfer (ZS-XLT): fine-tuned on sizable source-language task data, they perform the task in target languages without labeled instances. The effectiveness of ZS-XLT hinges on the linguistic proximity between languages and the amount of pretraining data for a language. Because of this, model selection based on source-language validation is unreliable: it picks model snapshots with suboptimal target-language performance. As a remedy, some work optimizes ZS-XLT by extensively tuning hyperparameters: the follow-up work then routinely struggles to replicate the original results. Other work searches over narrower hyperparameter grids, reporting substantially lower performance. In this work, we therefore propose an unsupervised evaluation protocol for ZS-XLT that decouples performance maximization from hyperparameter tuning. As a robust and more transparent alternative to extensive hyperparameter tuning, we propose to *accumulatively average* snapshots from different runs into a single model. We run broad ZS-XLT experiments on both higher-level semantic tasks (NLI, extractive QA) and a lower-level token classification task (NER) and find that conventional model selection based on source-language validation quickly plateaus to suboptimal ZS-XLT performance. On the other hand, our accumulative run-by-run averaging of models trained with different hyperparameters boosts ZS-XLT performance and closely correlates with "oracle" ZS-XLT, i.e., model selection based on target-language validation performance.

## 1 Introduction and Motivation

Massively multilingual transformers (MMTs) like XLM-{R,V} (Conneau et al., 2020; Liang et al., 2023) or mT5 (Xue et al., 2021) are pretrained via language modeling on vast corpora encompassing 100+ languages. MMT fine-tuned on labeled task data in a source language can transfer cross-lingually *zero-shot*, i.e. without further annotations, to target languages (Hu et al., 2020; Lauscher et al., 2020). However, pretraining corpora size and linguistic distance between the source and target language dictate the quality of XLT (Lauscher et al., 2020). This is why model selection based on source-language validation data unreliably correlates with ZS-XLT and selects checkpoints that yield suboptimal target-language performance (Keung et al., 2020). Worse yet, there is no "best practice" for replicating ZS-XLT results of prior work. Some works, as our results suggest (cf. §4), (1) exhaust extraordinarily large hyperparameter grids and (2) monitor target-language performance for the best transfer (i.e., violating "true" ZS-XLT) to outperform baselines (Conneau et al., 2020; Wei et al., 2021). Other works rerun baselines with little to no hyperparameter tuning (Hu et al., 2020; Wu and Dredze, 2020): the re-evaluation then often trails original results by non-negligible margins.[1] As a remedy, Keung et al. (2020) propose to evaluate ZS-XLT on the snapshot that generalizes best to validation data in the target language ("oracle" ZS-XLT): as such, oracle ZS-XLT stabilizes evaluation and denotes ideal transfer performance. Nonetheless, oracle ZS-XLT overstates the performance of true ZS-XLT, for which no target-language instances are available (Schmidt et al., 2023). If they are, target-language annotations are always better levered for training than for validation (Schmidt et al., 2022).

This calls for an evaluation protocol that (1) maximizes "true" ZS-XLT results and (2) makes them easily reproducible, regardless of the extent of hyperparameter tuning. In this work, we find that model averaging fulfills both criteria. Weights averaging has proven effective in, e.g., MT (Vaswani et al., 2017) and recently NLU (Wang et al., 2022; Schmidt et al., 2023). Schmidt et al. (2023) enable

---

[1] For instance, when evaluating XLM-V$_{base}$, Liang et al. (2023) have been unable to reproduce the original results of XLM-R$_{base}$ on the XNLI benchmark (Conneau et al., 2020).

model averaging for sizable gains in XLT. They first fine-tune an MMT on labeled source-language data and then re-train models (i.e., more runs) by copying and freezing the task head of the initially fine-tuned model: this aligns snapshots and enables weight averaging across runs.[2]

**Contributions.** In this work, we propose an evaluation protocol that decouples maximizing ZS-XLT performance from hyperparameter tuning. The key idea is to *accumulatively average* snapshots of runs with different hyperparameters: this improves performance over model selection based on source-language validation performance. We run exhaustive experiments on higher-level (NLI, extractive QA) and lower-level (NER) NLU tasks on a broad grid of hyperparameters and show, examining the cross-section of all runs, that model selection based on source-language validation almost exclusively picks snapshots suboptimal for ZS-XLT. We also confirm that conventional hyperparameter tuning on source-language validation prematurely settles for models that maximize source-language performance at the expense of ZS-XLT. Crucially, we show that accumulative model averaging performs on par or better than the best snapshot picked by source-language validation already from the second (i.e. first averaged-in) run and then consistently improves ZS-XLT with more runs. We additionally show that this accumulative model averaging closely correlates with oracle ZS-XLT *without* requiring any source- or target-language validation data to maximize transfer performance.

## 2 Accumulative Run Averaging

Prior work conducts model selection for ZS-XLT by extensive hyperparameter tuning using either source- or target-language validation data. Whereas the latter violates true ZS-XLT (Schmidt et al., 2022), the former overfits to source-language performance (Keung et al., 2020). The recent success of snapshot averaging in XLT (Schmidt et al., 2023) motivates our research question: can (accumulative) averaging of models trained during hyperparameter search outperform – with fewer overall training runs – the ZS-XLT performance of the "optimal" model selected based on source-language validation performance?

We benchmark model selection based on source-

language validation against accumulative model averaging as follows. We iteratively sample models (i.e., runs) $\{\{\theta_1, \ldots, \theta_r\} \mid 1 \leq r \leq 10\}$ with different hyperparameters (i.e., pairs of learning rates and batch sizes) from the pool of runs containing $N$ runs per hyperparameter configuration (cf. Appendix §A.2). We repeat this procedure 10 times and report mean ZS-XLT performance. Standard model selection picks the model $\{\arg\max_i \text{Val}(\theta_i) \mid 1 \leq i \leq r\}$ at run $r$ that maximizes source- (target-) language validation, capturing the "true" ("oracle") ZS-XLT performance. "Accumulative averaging", in contrast, naively averages (i.e. without any supervision) all models of $r$ runs to a single model $\frac{1}{r} \sum_{j=1}^{r} \theta_j = \bar{\theta}_r$.

## 3 Experimental Setup

**Tasks and Languages.** We select for our evaluation two higher-level semantic tasks (NLI and and extractive QA) and one lower-level structured prediction task (NER). For each task, we fine-tune the MMT on the provided English training splits.[3]

*Natural Language Inference* (NLI). We evaluate NLI on XNLI (Conneau et al., 2018) and IndicXNLI (Aggarwal et al., 2022) which together cover 25 typologically diverse languages.

*Extractive QA* (TyDiQA-GoldP). TyDiQA-GoldP comprises questions that are answered by a span of text in the provided gold passage and covers 9 diverse languages (Clark et al., 2020).

*Named Entity Recognition* (NER). We evaluate NER on 25 languages from WikiANN (Pan et al., 2017), 10 languages from MasakhaNER (Adelani et al., 2021), and 9 languages from MasakhaNER 2.0 (Adelani et al., 2022).

**Training Details.** We train XLM-R$_{\text{large}}$ (Conneau et al., 2020) for 10 epochs with AdamW (Loshchilov and Hutter, 2019), weight decay of $0.01$, gradient norm clipping to $1.0$, and a LR schedule of 10% linear warm-up and decay.[4] We save 10 snapshots per model, one at every 10% of total training steps. The maximum sequence length is 128 tokens for NLI and NER and 384 with a stride of 128 for TyDiQA-GoldP.

**Hyperparameter Grids.** We simulate conventional hyperparameter grid search over a broad set

---

[2]Fine-tuning models with different randomly initialized task heads otherwise yields sets of incompatible weights, hindering meaningful model averaging.

[3]Train portion of MNLI (Williams et al., 2018), the enclosed 3,696 English training instances of TyDiQA-GoldP for QA, and the English training portion of WikiANN for NER.

[4]The training data of TyDiQA-GoldP consists of merely 3,696 instances; we thus fine-tune longer, for 40 epochs.

of 21 configurations, pairing seven learning rates $l \in \{0.1, 0.5, 1, 1.5, 2, 2.5, 3\}e^{-5}$ with three batch sizes $b \in \{16, 32, 64\}$. The grid is deliberately kept wide and same for all tasks to not reflect any prior knowledge on task-specific "good values".[5] We retrain MMT for each pair $(l, b)$ three times with different random seeds to account for variances over individual runs.

**Model Variants.** We evaluate four model variants: $v \in \{\text{LAST}, \text{SRC-DEV}, \text{CA}, \text{TRG-DEV}\}$. LAST is simply the final snapshot of a training run. SRC-DEV is the snapshot that maximizes source-language validation performance (Hu et al., 2020). CA averages all snapshots of a run to a single model and, according to Schmidt et al. (2023), outperforms LAST and SRC-DEV. TRG-DEV breaches "true" ZS-XLT and picks the snapshot that performs best on the target-language validation data (Keung et al., 2020): as such, it generally represents an upper-bound of single-run ZS-XLT performance.[6]

## 4 Results and Discussion

**Single-Run Performance.** The full ZS-XLT results by hyperparameters are presented in Appendix §A.2 (cf. Table 3). We observe that optimal ZS-XLT of single runs depends on all axes of analysis: task, hyperparameters, and model variant. While LAST and SRC-DEV generally perform well, their ZS-XLT performance fluctuates substantially across hyperparameter configurations, in line with (Keung et al., 2020; Schmidt et al., 2023). CA is a strong and robust baseline that often outperforms LAST and SRC-DEV by notable margins on TyDiQA and NER. In the context of a single run, CA performs especially well with suboptimal hyperparameters, even sometimes outperforming TRG-DEV. We also confirm that CA remedies variation in ZS-XLT both within and across hyperparameters (Schmidt et al., 2023). Table 3 (cf. Appendix §A.2) further highlights the notable gap in ZS-XLT performance between the best-performing hyperparameter configurations and those selected based on source-language validation. Only the "oracle" model selection based on target-language validation reliably correlates with the actual best (test) ZS-XLT performance.

**Run-by-Run Analysis.** Table 1 compares ZS-XLT performance run-for-run of all variants for model selection based on source-language validation (Max. SRC-DEV) against our accumulative averaging of randomly sampled runs with different hyperparameters (cf. §2). On NLI, picking a single model on source-language validation only improves ZS-XLT when moving from having one to having two models (i.e., between first two rows of Table 1) and stagnates when having more models to choose from. With more runs, source-language validation may even prefer models that are worse at ZS-XLT on TyDiQA and NER. Conventional model selection thus maximizes source-language performance at the expense of ZS-XLT. Across the board, accumulative averaging already matches or surpasses Max. SRC-DEV (with any number of models) using merely two or three runs. Moreover, accumulative averaging consistently outperforms the *overall best* single-run model chosen from $3+$ runs (highlighted in green), irrespective of the task. On all tasks, accumulative averaging stabilizes ZS-XLT and reduces performance variance vis-a-vis Max. SRC-DEV counterparts.

Accumulatively averaging within-run snapshots (CA) outperforms LAST and SRC-DEV slightly on NLI and materially on NER. For NER, ZS-XLT from WikiANN to MasakhaNER (2.0) also represents a domain transfer (from Wiki to news), in which CA yields tremendous gains. In-domain (i.e., test on WikiANN), CA generally performs on par with LAST and SRC-DEV. The same is not true for QA, where CA performs slightly worse: we ascribe this to averaging of "unconverged" snapshots, owing to the small TyDiQA training set (merely 3,696 instances), especially from runs with smaller learning rates and larger batches (cf. Table 3).

**Further Analyses.** Table 2 extends the run-by-run analysis to TRG-DEV and "model soups" (SOUP) to illustrate why accumulative model averaging outperforms model selection based on source-language validation. Rather than selecting a single snapshot, SOUP averages the five snapshots (among all available runs) with best source-language validation performance (Wortsman et al., 2022).

Compared to (oracle) TRG-DEV, accumulatively averaging runs performs on par on NLI, slightly better on TyDiQA, and somewhat worse on NER. TRG-DEV selects language-specific snapshots, thereby tailoring ZS-XLT to each target language and remedying for the varying performance of Max. SRC-DEV in ZS-XLT to *many* target languages. Such

---

[5] Our full results in Table 3 indicate that for each task we obtain maximal (oracle) ZS-XLT performance with a different, task-specific hyperparameter configurations.

[6] Irrespective of tasks and language, labeled instances in the target-language bring larger gains if used for training rather than for model selection (Schmidt et al., 2022).

| | NLI | | | | | | TyDiQA-GoldP | | | | | | NER | | | | | |
|---|---|---|---|---|---|---|---|---|---|---|---|---|---|---|---|---|---|---|
| | Max. SRC-DEV | | | Accumulative Averaging | | | Max. SRC-DEV | | | Accumulative Averaging | | | Max. SRC-DEV | | | Accumulative Averaging | | |
| $r$ | LAST | S-DEV | CA | LAST | S-DEV | CA | LAST | S-DEV | CA | LAST | S-DEV | CA | LAST | S-DEV | CA | LAST | S-DEV | CA |
| 1 | $76.5_{0.6}$ | $76.5_{0.8}$ | $77.3_{0.4}$ | $76.5_{0.6}$ | $76.5_{0.8}$ | $77.3_{0.4}$ | $71.9_{0.4}$ | $71.9_{0.7}$ | $73.6_{1.9}$ | $71.9_{0.4}$ | $71.9_{0.7}$ | $73.6_{1.9}$ | $\mathbf{40.8_{2.7}}$ | $41.1_{3.1}$ | $\mathbf{44.6_{2.1}}$ | $40.8_{2.7}$ | $41.1_{3.0}$ | $44.6_{2.1}$ |
| 2 | $77.2_{0.3}$ | $77.5_{0.4}$ | $\mathbf{77.6_{0.2}}$ | $77.6_{0.3}$ | $77.8_{0.4}$ | $78.0_{0.2}$ | $71.9_{0.6}$ | $71.6_{0.6}$ | $73.3_{2.0}$ | $73.4_{1.2}$ | $73.3_{1.1}$ | $72.9_{2.8}$ | $39.3_{2.1}$ | $39.3_{2.1}$ | $43.5_{1.1}$ | $43.2_{2.2}$ | $43.2_{2.2}$ | $45.6_{1.5}$ |
| 3 | $77.2_{0.3}$ | $77.5_{0.4}$ | $77.6_{0.2}$ | $77.8_{0.3}$ | $77.9_{0.4}$ | $78.1_{0.2}$ | $72.1_{0.8}$ | $71.8_{0.8}$ | $74.1_{1.0}$ | $74.1_{0.7}$ | $74.2_{0.7}$ | $73.8_{1.2}$ | $39.3_{1.2}$ | $39.5_{1.7}$ | $44.0_{1.1}$ | $45.0_{1.7}$ | $45.1_{1.8}$ | $47.3_{1.3}$ |
| 4 | $77.2_{0.4}$ | $77.5_{0.4}$ | $77.5_{0.2}$ | $77.7_{0.3}$ | $77.9_{0.4}$ | $78.1_{0.3}$ | $72.5_{0.8}$ | $\mathbf{72.0_{0.9}}$ | $73.9_{0.6}$ | $74.5_{0.6}$ | $74.1_{0.4}$ | $74.1_{0.9}$ | $40.2_{2.0}$ | $40.8_{2.2}$ | $44.5_{1.5}$ | $45.0_{1.7}$ | $45.3_{1.8}$ | $47.2_{1.4}$ |
| 5 | $77.3_{0.3}$ | $\mathbf{77.6_{0.3}}$ | $77.5_{0.2}$ | $77.9_{0.2}$ | $78.0_{0.2}$ | $78.1_{0.1}$ | $\mathbf{72.6_{0.8}}$ | $72.0_{0.9}$ | $73.8_{0.8}$ | $\mathbf{74.7_{0.7}}$ | $\mathbf{74.4_{0.6}}$ | $74.2_{0.8}$ | $40.3_{2.0}$ | $\mathbf{41.2_{2.3}}$ | $43.9_{1.9}$ | $45.3_{1.7}$ | $45.5_{1.7}$ | $47.5_{1.4}$ |
| 6 | $77.3_{0.3}$ | $77.6_{0.1}$ | $77.5_{0.2}$ | $77.9_{0.1}$ | $78.0_{0.2}$ | $78.1_{0.1}$ | $72.6_{0.8}$ | $72.0_{0.9}$ | $74.2_{0.5}$ | $74.7_{0.7}$ | $74.4_{0.5}$ | $74.2_{0.7}$ | $40.3_{2.0}$ | $41.2_{2.2}$ | $43.9_{1.9}$ | $45.7_{1.4}$ | $45.9_{1.4}$ | $47.9_{1.2}$ |
| 7 | $77.3_{0.3}$ | $77.6_{0.1}$ | $77.5_{0.2}$ | $77.9_{0.2}$ | $78.1_{0.2}$ | $78.2_{0.2}$ | $72.3_{0.9}$ | $71.7_{0.7}$ | $\mathbf{74.3_{0.3}}$ | $74.6_{0.7}$ | $74.3_{0.6}$ | $74.2_{0.5}$ | $40.0_{2.1}$ | $40.6_{2.3}$ | $44.1_{2.0}$ | $46.0_{1.3}$ | $46.1_{1.3}$ | $48.1_{1.1}$ |
| 8 | $77.3_{0.3}$ | $77.6_{0.2}$ | $77.5_{0.2}$ | $\mathbf{78.0_{0.2}}$ | $\mathbf{78.2_{0.1}}$ | $\underline{\mathbf{78.3_{0.2}}}$ | $72.1_{0.9}$ | $71.7_{0.7}$ | $74.2_{0.5}$ | $74.6_{0.7}$ | $74.3_{0.5}$ | $\mathbf{74.3_{0.5}}$ | $40.0_{2.1}$ | $40.6_{2.3}$ | $44.6_{1.7}$ | $46.0_{1.1}$ | $46.1_{1.2}$ | $48.2_{1.0}$ |
| 9 | $\mathbf{77.4_{0.2}}$ | $77.6_{0.2}$ | $77.6_{0.2}$ | $78.0_{0.1}$ | $78.1_{0.1}$ | $78.3_{0.2}$ | $72.0_{1.1}$ | $71.7_{0.7}$ | $74.2_{0.5}$ | $74.6_{0.5}$ | $74.4_{0.4}$ | $74.2_{0.4}$ | $39.6_{2.3}$ | $39.9_{2.4}$ | $44.3_{1.8}$ | $46.0_{0.6}$ | $46.1_{0.7}$ | $48.3_{0.7}$ |
| 10 | $77.3_{0.2}$ | $77.6_{0.2}$ | $77.6_{0.2}$ | $78.0_{0.1}$ | $78.2_{0.1}$ | $78.3_{0.1}$ | $72.0_{1.1}$ | $71.7_{0.7}$ | $74.2_{0.5}$ | $74.6_{0.6}$ | $74.4_{0.4}$ | $74.2_{0.6}$ | $39.6_{2.3}$ | $39.9_{2.4}$ | $44.4_{1.7}$ | $\mathbf{46.1_{0.5}}$ | $\mathbf{46.2_{0.6}}$ | $\underline{\mathbf{48.4_{0.5}}}$ |

Table 1: $\{\{\theta_1, \ldots, \theta_r\} \mid 1 \le r \le 10\}$ models sampled for variants $v \in \{\text{LAST, SRC-DEV, CA}\}$ from Table 3 (cf. §3). "Max. SRC-DEV" picks the run $\{\arg\max_i \text{SrcVal}(\theta_i^v) \mid 1 \le i \le r\}$. "Accumulative averaging" simply averages all runs $\frac{1}{r}\sum_{j=1}^r \theta_j^v$. Metrics: accuracy for NLI, span-$F_1$ for TyDiQA and token-level $F_1$ for NER. Subscripts denote std. deviation. Colored averaging **outperforms +0.2 or more** or **performs $\pm0.1$ of** the best Max. SRC-DEV model.

| | NLI | | | | TyDiQA-GoldP | | | | NER | | | |
|---|---|---|---|---|---|---|---|---|---|---|---|---|
| | Max. Dev | | Acc. Avg. | | Max. Dev | | Acc. Avg. | | Max. Dev | | Acc. Avg. | |
| $r$ | SRC DEV | TRG DEV | CA | SOUP | SRC DEV | TRG DEV | CA | SOUP | SRC DEV | TRG DEV | CA | SOUP |
| 1 | 77.3 | 77.0 | 77.3 | 76.8 | 71.9 | 72.8 | 73.6 | 73.7 | 41.1 | 46.5 | 44.6 | 42.3 |
| 3 | 77.5 | 77.7 | 78.1 | 77.6 | 71.8 | 73.5 | 73.8 | 73.8 | 39.5 | 49.2 | 47.3 | 42.1 |
| 5 | **77.6** | 77.9 | 78.1 | 77.6 | **72.0** | 73.4 | **74.2** | **74.3** | **41.2** | 49.7 | 47.5 | **42.8** |
| 7 | 77.6 | 78.2 | 78.2 | **77.8** | 71.7 | 73.7 | 74.2 | 73.9 | 40.6 | **49.9** | 48.1 | 42.8 |
| 10 | 77.6 | **78.4** | **78.3** | 77.7 | 71.7 | **73.9** | 74.2 | 73.8 | 39.9 | 49.9 | **48.4** | 42.8 |

Table 2: "Max. TRG-DEV" selects the run $\{\arg\max_i \frac{1}{|T|}\text{Val}_T(\theta_i) \mid 1 \le i \le r\}$, where $T$ is the set of target languages. SOUP averages the five checkpoints (from all available runs) that "Max. SRC-DEV". For other details, see Table 1.

a variation has been shown to be particularly pronounced in ZS-XLT on token-level tasks like NER or POS (Schmidt et al., 2023). On TyDiQA, we believe that accumulative averaging (slightly) better stabilizes the transfer from a small training set (3.7K instances). SOUPs however perform notably worse than both TRG-DEV and accumulative averaging on NLI and NER. SOUPs lack the beneficial diversity of different runs, as the best snapshots often come from the same "good" run.[7] Anecdotal evidence further exemplifies why source-language validation is inapt for ZS-XLT. One of 63 SRC-DEV models replicates XNLI results of Conneau et al. (2020), vastly exceeding all other runs (c.$\Delta$+1.0). This "miraculous" run though merely ranks 3rd according to source-language validation performance.

The above suggests that even the more sophisticated hyperparameter tuning strategies (e.g., Bayesian optimization) are unlikely to improve ZS-

XLT without target-language validation. On the other hand, accumulative averaging improves ZS-XLT threefold: **(1)** Unlike model selection, it does not plateau in ZS-XLT on suboptimal single runs that maximize source-language performance; **(2)** TRG-DEV showcases that accumulative averaging ingests further runs with snapshots that perform well on ZS-XLT; **(3)** Model averaging irons out idiosyncratic noise of individual runs, leading to better performance. This renders accumulative averaging a robust (i.e., replicable results) and fair (i.e. true zero-shot) evaluation protocol for ZS-XLT.

## 5 Conclusion

Inconsistent hyperparameter tuning and model selection protocols exacerbate replicating previous results on ZS-XLT. In this focused study, we devise a ZS-XLT evaluation protocol that addresses previous shortcomings and feeds two birds with one scone. We show that accumulatively averaging snapshots – rather than selecting models based on source-language validation performance – both improves and stabilizes ZS-XLT. Conventional model selection strategies prematurely settle for models that maximize source-language validation performance and discard runs that generalize better in ZS-XLT. Accumulative model averaging both incorporates snapshots that transfer well and irons out models that perform badly. We find that model averaging correlates closely with "oracle" ZS-XLT, which assumes models selection on target-language validation instances. We hope future work adopts model averaging to promote fair and reproducible ZS-XLT that puts models on equal footing.

---

[7]Extending SOUP to average the top-10 best snapshots does not improve performance.

## Limitations

Additional factors must be taken into consideration, even though we aspire to evaluate ZS-XLT on all levels of transparency (i.e., variants and strategies) across a varied set of downstream tasks on broad hyperparameter grids. Neither model selection on source-language validation data nor accumulative averaging may benefit ZS-XLT on certain tasks, as Schmidt et al. (2023), e.g., do not find that any variant other than TRG-DEV yields gain over LAST on part-of-speech tagging. The underlying cause remains unclear. For instance, the gains on ZS-XLT stemming from model selection or accumulative averaging likely depend on the type of distributional shift from the source-language training data and the target-language instances to transfer to (cf. §4; e.g. dynamics of variants in ZS-XLT for NER). accumulative averaging nevertheless remains a robust evaluation protocol, as ZS-XLT performance is not expected to deteriorate via-à-vis other "fair" strategies (e.g., max. SRC-DEV). In addition, there may exist a subset of pairs of learning rates and batch sizes that jointly maximize source- and target-language performance. However, as our results suggest (§4), runs on such hyperparameters likely are indistinguishable from those that exclusively perform just as well on the source-language validation set.

## Acknowledgments

We thank the state of Baden-Württemberg for its support through access to the bwHPC. Ivan Vulić is supported by a personal Royal Society University Research Fellowship 'Inclusive and Sustainable Language Technology for a Truly Multilingual World' (no 221137; 2022–).

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

# A Appendix

## A.1 Reproduction Details

**Code**. Our code is available at: https://github.com/fdschmidt93/ofa-xlt

**Model architectures.** All models use the `AutoModelFor{SequenceClassification, TokenClassification, QuestionAnswering}` of `xlm-roberta-large` for the corresponding task from the `transformers` library (Wolf et al., 2020).

**Compute Requirements.** We execute all experiments on a single V100 with 32GB VRAM. We estimate that we require total compute time of c.1,050 hours over all fine-tuning iterations and evaluations. We arrive at this budget as follows. We on average train models on NLI for about 11.5 hours, on TyDiQA-GoldP for roughly 1.5 hours, and on NER for an estimated 3 hours. We therefore execute 63 training runs (21 hyperparameter configurations ran on for 3 seeds, cf. §3) for 16 hours for a total of c.1K GPU hours. We loosely estimate that accumulative averaging adds another 50 hours of runtime for evaluation.

**Model Averaging.** We follow Schmidt et al. (2023) to enabling accumulative averaging of checkpoints for NLI and TyDiQA-GoldP. For these tasks, we initially fine-tune XLM-R$_{large}$ with a batch size of 32 and a learning rate of $2e^{-5}$. For NER, we find that merely randomly initializing the tasks heads across all runs with the same head slightly improves performance ( $\Delta + 1.0$) of all variants in single-run and accumulative averaging. We suspect that the original language modelling weights better align with NER as a token-level classification task and do not diverge to incompatible sets of parameters in fine-tuning (cf. §3 of Schmidt et al. (2023)).

## A.2 Full Results By Hyperparameter Configuration

**ZS-XLT Performance**

| Hyperparameters | | NLI | | | | TyDiQA-GoldP | | | | NER | | | |
|---|---|---|---|---|---|---|---|---|---|---|---|---|---|
| Learning Rate | Batch Size | LAST | SRC-DEV | CA | TRG-DEV | LAST | SRC-DEV | CA | TRG-DEV | LAST | SRC-DEV | CA | TRG-DEV |
| $1e^{-6}$ | 16 | $77.2_{0.1}$ | $77.5_{0.3}$ | $77.6_{0.2}$ | $77.9_{0.5}$ | $71.3_{0.3}$ | $71.3_{0.1}$ | $68.4_{0.8}$ | $71.4_{0.3}$ | $\mathbf{45.9_{0.3}}$ | $\mathbf{45.9_{0.3}}$ | $\mathbf{46.5_{0.3}}$ | $47.8_{0.1}$ |
| | 32 | $\mathbf{77.4_{0.1}}$ | $77.5_{0.2}$ | $77.7_{0.2}$ | $78.1_{0.1}$ | $71.2_{0.2}$ | $71.3_{0.4}$ | $63.2_{0.3}$ | $71.6_{0.2}$ | $45.3_{0.2}$ | $45.3_{0.2}$ | $44.7_{0.3}$ | $45.8_{0.3}$ |
| | 64 | $77.6_{0.1}$ | $77.6_{0.3}$ | $77.4_{0.1}$ | $77.8_{0.1}$ | $70.9_{0.4}$ | $70.9_{0.4}$ | $49.8_{0.2}$ | $70.9_{0.1}$ | $45.3_{0.2}$ | $45.3_{0.2}$ | $42.6_{0.3}$ | $45.5_{0.2}$ |
| $5e^{-6}$ | 16 | $76.7_{0.2}$ | $76.9_{0.2}$ | $77.5_{0.3}$ | $77.7_{1.1}$ | $71.6_{0.2}$ | $71.9_{0.7}$ | $72.2_{0.1}$ | $72.0_{0.4}$ | $44.0_{0.9}$ | $44.9_{0.9}$ | $47.4_{0.3}$ | $49.8_{0.3}$ |
| | 32 | $76.8_{0.0}$ | $77.6_{0.4}$ | $77.6_{0.1}$ | $78.3_{0.5}$ | $71.6_{0.1}$ | $71.5_{0.1}$ | $71.6_{0.1}$ | $72.3_{0.4}$ | $42.8_{0.3}$ | $42.9_{0.2}$ | $45.8_{0.6}$ | $47.7_{1.0}$ |
| | 64 | $77.0_{0.2}$ | $\mathbf{78.1_{0.6}}$ | $\mathbf{77.8_{0.2}}$ | $\mathbf{78.3_{0.3}}$ | $71.0_{0.9}$ | $70.9_{0.6}$ | $69.0_{0.7}$ | $71.7_{0.1}$ | $43.8_{1.6}$ | $43.8_{1.6}$ | $46.2_{1.3}$ | $49.0_{0.4}$ |
| $1e^{-5}$ | 16 | $76.6_{0.2}$ | $76.6_{0.2}$ | $77.5_{0.2}$ | $77.1_{0.2}$ | $73.0_{0.4}$ | $72.5_{0.4}$ | $73.5_{0.5}$ | $73.7_{0.6}$ | $40.6_{0.1}$ | $40.7_{1.8}$ | $43.9_{1.4}$ | $48.0_{2.5}$ |
| | 32 | $76.8_{0.2}$ | $77.1_{0.4}$ | $77.6_{0.1}$ | $77.2_{0.2}$ | $72.1_{0.4}$ | $72.6_{0.5}$ | $73.0_{0.3}$ | $73.2_{0.3}$ | $40.1_{1.8}$ | $40.1_{1.8}$ | $42.8_{1.5}$ | $46.0_{3.1}$ |
| | 64 | $76.8_{0.3}$ | $77.2_{0.5}$ | $77.5_{0.2}$ | $77.7_{0.2}$ | $72.0_{0.8}$ | $71.7_{0.8}$ | $71.7_{0.2}$ | $73.0_{0.3}$ | $43.1_{1.9}$ | $43.4_{2.3}$ | $46.6_{1.2}$ | $49.7_{0.7}$ |
| $1.5e^{-5}$ | 16 | $75.6_{0.1}$ | $75.6_{0.2}$ | $76.8_{0.3}$ | $76.5_{0.4}$ | $73.7_{0.5}$ | $\mathbf{73.3_{0.3}}$ | $74.4_{0.6}$ | $\mathbf{74.1_{0.4}}$ | $41.0_{3.1}$ | $42.0_{2.6}$ | $45.3_{2.0}$ | $\mathbf{50.0_{1.1}}$ |
| | 32 | $76.6_{0.1}$ | $76.5_{0.1}$ | $77.4_{0.2}$ | $77.0_{0.1}$ | $73.0_{0.7}$ | $73.1_{0.9}$ | $74.0_{0.1}$ | $73.7_{0.6}$ | $39.9_{1.1}$ | $40.6_{1.3}$ | $43.3_{0.2}$ | $46.2_{1.9}$ |
| | 64 | $76.8_{0.1}$ | $77.1_{0.5}$ | $77.6_{0.3}$ | $77.7_{0.6}$ | $72.9_{0.5}$ | $72.6_{0.7}$ | $73.3_{0.4}$ | $73.5_{0.3}$ | $40.6_{2.1}$ | $41.1_{2.0}$ | $42.8_{1.5}$ | $46.0_{1.1}$ |
| $2e^{-5}$ | 16 | $74.1_{0.3}$ | $74.1_{0.3}$ | $76.1_{0.2}$ | $74.8_{0.2}$ | $72.9_{0.5}$ | $72.7_{0.2}$ | $74.1_{0.2}$ | $73.5_{0.2}$ | $38.6_{1.2}$ | $38.6_{1.2}$ | $43.8_{1.4}$ | $46.2_{3.0}$ |
| | 32 | $75.9_{0.4}$ | $76.1_{0.3}$ | $77.1_{0.1}$ | $76.4_{0.5}$ | $73.1_{0.1}$ | $72.3_{0.9}$ | $74.1_{0.2}$ | $73.3_{0.4}$ | $39.3_{0.7}$ | $39.0_{1.1}$ | $42.3_{1.5}$ | $44.3_{2.5}$ |
| | 64 | $76.6_{0.5}$ | $77.0_{0.4}$ | $77.4_{0.1}$ | $77.7_{0.2}$ | $71.9_{0.4}$ | $71.7_{1.1}$ | $73.3_{0.5}$ | $72.8_{0.4}$ | $39.5_{1.2}$ | $39.7_{1.3}$ | $42.3_{1.1}$ | $46.2_{0.4}$ |
| $2.5e^{-5}$ | 16 | $71.5_{0.2}$ | $71.3_{0.4}$ | $75.0_{0.2}$ | $73.1_{0.1}$ | $\mathbf{73.2_{0.4}}$ | $72.4_{0.9}$ | $74.7_{0.1}$ | $73.4_{0.5}$ | $39.3_{1.0}$ | $39.3_{1.0}$ | $44.3_{1.7}$ | $47.1_{0.6}$ |
| | 32 | $74.8_{0.2}$ | $74.8_{0.2}$ | $76.3_{0.1}$ | $76.1_{0.2}$ | $72.3_{0.2}$ | $71.1_{0.4}$ | $74.8_{0.3}$ | $73.4_{0.3}$ | $39.4_{1.5}$ | $39.4_{1.5}$ | $42.5_{0.6}$ | $45.0_{0.7}$ |
| | 64 | $76.7_{0.2}$ | $76.7_{0.2}$ | $77.5_{0.1}$ | $77.3_{0.5}$ | $72.6_{0.6}$ | $72.3_{0.7}$ | $74.2_{0.1}$ | $73.3_{1.0}$ | $39.9_{1.0}$ | $39.9_{1.0}$ | $43.5_{1.5}$ | $47.7_{2.3}$ |
| $3e^{-5}$ | 16 | $67.9_{0.6}$ | $67.7_{0.2}$ | $73.0_{0.3}$ | $70.8_{0.9}$ | $72.3_{0.2}$ | $71.7_{0.4}$ | $74.4_{0.4}$ | $72.4_{0.5}$ | $37.4_{1.5}$ | $37.3_{1.1}$ | $43.0_{0.8}$ | $46.3_{2.5}$ |
| | 32 | $73.3_{0.1}$ | $73.3_{0.4}$ | $75.8_{0.2}$ | $74.4_{0.1}$ | $71.7_{0.4}$ | $71.6_{0.7}$ | $\mathbf{75.1_{0.1}}$ | $73.6_{0.8}$ | $37.9_{1.6}$ | $38.0_{1.8}$ | $43.7_{1.6}$ | $47.2_{2.7}$ |
| | 64 | $75.6_{0.1}$ | $75.4_{0.3}$ | $76.8_{0.3}$ | $76.3_{0.7}$ | $72.0_{0.2}$ | $71.8_{0.4}$ | $74.1_{0.5}$ | $73.2_{0.4}$ | $39.0_{1.9}$ | $39.4_{1.4}$ | $42.1_{1.2}$ | $44.0_{1.2}$ |
| | $\Delta$ | 0.0 | 0.6 | 0.3 | 0.0 | 2.0 | 2.0 | 1.0 | 0.0 | 4.9 | 5.2 | 2.6 | 0.2 |

Table 3: ZS-XLT averaged over all target languages by task, model variant, and hyperparameters (cf. §3). For each column, **best ZS-XLT emphasized in bold** and max. validation performance (cf. Table 4) shaded in green. $\Delta$ is the difference of **best ZS-XLT** and ZS-XLT on models that maximize validation performance. **Metrics:** accuracy for NLI, span-$F_1$ for TyDiQA and token-level $F_1$ for NER. Subscripts denote std. deviation.

**Validation Set Performance**

| Hyperparameters | | NLI | | | | TyDiQA-GoldP | | | | NER | | | |
|---|---|---|---|---|---|---|---|---|---|---|---|---|---|
| Learning Rate | Batch Size | LAST | SRC-DEV | CA | TRG-DEV | LAST | SRC-DEV | CA | TRG-DEV | LAST | SRC-DEV | CA | TRG-DEV |
| $1e^{-6}$ | 16 | $\mathbf{90.2_{0.2}}$ | $\mathbf{90.5_{0.2}}$ | $90.1_{0.2}$ | $79.2_{0.3}$ | $76.4_{0.2}$ | $76.7_{0.5}$ | $75.1_{0.3}$ | $67.0_{0.2}$ | $81.9_{0.1}$ | $81.9_{0.1}$ | $79.5_{0.2}$ | $49.9_{0.1}$ |
| | 32 | $\mathbf{90.2_{0.2}}$ | $90.3_{0.1}$ | $90.0_{0.0}$ | $79.2_{0.1}$ | $\mathbf{77.0_{0.5}}$ | $\mathbf{77.9_{0.8}}$ | $73.5_{0.4}$ | $66.9_{0.1}$ | $80.4_{0.1}$ | $80.4_{0.1}$ | $76.5_{0.3}$ | $48.3_{0.3}$ |
| | 64 | $90.1_{0.1}$ | $90.2_{0.1}$ | $89.6_{0.1}$ | $78.9_{0.0}$ | $76.0_{0.6}$ | $76.3_{0.4}$ | $64.4_{0.2}$ | $66.5_{0.2}$ | $78.0_{0.1}$ | $78.0_{0.1}$ | $71.4_{0.4}$ | $47.7_{0.2}$ |
| $5e^{-6}$ | 16 | $89.3_{0.4}$ | $89.7_{0.2}$ | $90.1_{0.2}$ | $78.9_{0.6}$ | $73.8_{0.9}$ | $76.0_{0.5}$ | $75.9_{0.8}$ | $68.7_{0.1}$ | $85.1_{0.3}$ | $85.3_{0.4}$ | $84.8_{0.1}$ | $\mathbf{52.1_{0.4}}$ |
| | 32 | $89.5_{0.4}$ | $89.9_{0.2}$ | $90.1_{0.2}$ | $79.4_{0.4}$ | $74.5_{0.5}$ | $75.8_{0.7}$ | $76.2_{0.8}$ | $68.2_{0.2}$ | $84.7_{0.1}$ | $84.7_{0.1}$ | $84.0_{0.2}$ | $50.0_{1.2}$ |
| | 64 | $89.6_{0.1}$ | $90.2_{0.3}$ | $90.0_{0.1}$ | $\mathbf{79.4_{0.3}}$ | $75.1_{0.7}$ | $76.3_{0.4}$ | $75.1_{0.2}$ | $67.8_{0.2}$ | $84.2_{0.2}$ | $84.2_{0.2}$ | $83.0_{0.3}$ | $51.2_{0.4}$ |
| $1e^{-5}$ | 16 | $88.9_{0.3}$ | $89.1_{0.2}$ | $89.6_{0.1}$ | $78.2_{0.1}$ | $74.5_{0.4}$ | $75.7_{0.1}$ | $75.8_{0.6}$ | $69.4_{0.3}$ | $85.6_{0.1}$ | $\mathbf{85.7_{0.1}}$ | $85.8_{0.2}$ | $50.3_{2.2}$ |
| | 32 | $89.1_{0.1}$ | $89.3_{0.2}$ | $89.6_{0.1}$ | $78.6_{0.3}$ | $74.4_{0.6}$ | $75.9_{0.6}$ | $75.7_{1.0}$ | $69.2_{0.1}$ | $85.4_{0.1}$ | $85.5_{0.2}$ | $85.3_{0.2}$ | $48.4_{3.1}$ |
| | 64 | $89.7_{0.4}$ | $89.9_{0.0}$ | $\mathbf{90.1_{0.1}}$ | $79.1_{0.3}$ | $74.8_{1.2}$ | $76.1_{0.0}$ | $76.7_{0.4}$ | $69.0_{0.2}$ | $85.0_{0.1}$ | $85.0_{0.1}$ | $84.5_{0.1}$ | $52.0_{0.8}$ |
| $1.5e^{-5}$ | 16 | $88.4_{0.3}$ | $88.5_{0.2}$ | $89.1_{0.2}$ | $77.5_{0.2}$ | $74.8_{0.7}$ | $76.3_{0.3}$ | $76.5_{0.7}$ | $\mathbf{69.7_{0.2}}$ | $\mathbf{86.0_{0.1}}$ | $86.1_{0.1}$ | $86.2_{0.1}$ | $52.0_{1.3}$ |
| | 32 | $89.0_{0.5}$ | $89.0_{0.5}$ | $89.6_{0.1}$ | $78.2_{0.2}$ | $75.1_{0.7}$ | $76.5_{0.4}$ | $76.5_{0.6}$ | $69.5_{0.4}$ | $85.3_{0.2}$ | $85.4_{0.2}$ | $85.3_{0.2}$ | $48.3_{1.6}$ |
| | 64 | $89.0_{0.3}$ | $89.4_{0.3}$ | $89.5_{0.2}$ | $78.7_{0.3}$ | $75.5_{0.9}$ | $76.2_{0.3}$ | $76.3_{0.8}$ | $69.3_{0.2}$ | $85.1_{0.2}$ | $85.2_{0.1}$ | $85.2_{0.3}$ | $48.3_{1.1}$ |
| $2e^{-5}$ | 16 | $87.7_{0.6}$ | $87.9_{0.3}$ | $88.9_{0.4}$ | $75.8_{0.2}$ | $74.6_{0.7}$ | $76.5_{0.8}$ | $76.8_{0.7}$ | $69.3_{0.1}$ | $85.9_{0.2}$ | $85.9_{0.2}$ | $86.0_{0.2}$ | $48.2_{3.1}$ |
| | 32 | $88.7_{0.2}$ | $88.8_{0.1}$ | $89.3_{0.4}$ | $77.6_{0.3}$ | $76.5_{1.5}$ | $77.1_{1.0}$ | $\mathbf{78.1_{0.5}}$ | $69.4_{0.5}$ | $85.4_{0.2}$ | $85.4_{0.2}$ | $85.7_{0.2}$ | $46.8_{2.5}$ |
| | 64 | $89.3_{0.2}$ | $89.4_{0.1}$ | $89.8_{0.1}$ | $78.7_{0.2}$ | $73.9_{0.7}$ | $76.3_{0.3}$ | $76.3_{0.3}$ | $69.2_{0.1}$ | $85.2_{0.2}$ | $85.4_{0.3}$ | $85.6_{0.3}$ | $48.4_{0.6}$ |
| $2.5e^{-5}$ | 16 | $87.5_{0.3}$ | $87.7_{0.1}$ | $88.6_{0.3}$ | $74.0_{0.3}$ | $75.8_{0.9}$ | $76.4_{0.2}$ | $77.2_{0.5}$ | $69.3_{0.3}$ | $85.5_{0.2}$ | $85.5_{0.2}$ | $86.0_{0.3}$ | $49.5_{0.5}$ |
| | 32 | $88.6_{0.1}$ | $88.6_{0.1}$ | $89.0_{0.2}$ | $76.9_{0.4}$ | $74.7_{0.6}$ | $76.0_{0.3}$ | $77.0_{0.6}$ | $69.1_{0.1}$ | $85.7_{0.2}$ | $85.7_{0.2}$ | $86.0_{0.1}$ | $46.9_{0.6}$ |
| | 64 | $88.8_{0.3}$ | $88.9_{0.2}$ | $89.4_{0.2}$ | $78.4_{0.4}$ | $75.1_{1.8}$ | $76.1_{0.7}$ | $76.1_{0.2}$ | $69.2_{0.3}$ | $85.4_{0.0}$ | $85.4_{0.0}$ | $85.7_{0.3}$ | $49.8_{2.3}$ |
| $3e^{-5}$ | 16 | $86.7_{0.2}$ | $86.8_{0.2}$ | $87.7_{0.1}$ | $71.7_{0.7}$ | $74.1_{0.6}$ | $75.6_{1.5}$ | $76.1_{0.8}$ | $68.5_{0.2}$ | $85.5_{0.1}$ | $85.6_{0.1}$ | $86.2_{0.0}$ | $48.5_{2.4}$ |
| | 32 | $87.8_{0.3}$ | $88.0_{0.5}$ | $89.0_{0.3}$ | $75.3_{0.3}$ | $73.8_{1.3}$ | $76.2_{1.0}$ | $76.5_{0.8}$ | $69.1_{0.2}$ | $85.4_{0.1}$ | $85.4_{0.1}$ | $86.0_{0.1}$ | $49.4_{3.0}$ |
| | 64 | $88.4_{0.2}$ | $88.5_{0.1}$ | $89.4_{0.2}$ | $77.4_{0.7}$ | $74.8_{0.6}$ | $76.3_{0.2}$ | $77.7_{0.4}$ | $69.1_{0.2}$ | $85.4_{0.1}$ | $85.4_{0.1}$ | $85.8_{0.1}$ | $46.2_{1.5}$ |

Table 4: Validation performance by task, model variant, and hyperparameters (cf. §3). LAST, SRC-DEV, and CA validate on source-language validation splits; TRG-DEV denotes performance averaged over individual snapshots of a run that perform best by target-language validation set. For each column, **best validation performance in bold**. **Metrics:** accuracy for NLI, span-$F_1$ for TyDiQA and token-level $F_1$ for NER. Subscripts denote std. deviation.