# OpenReview forum: "One For All $\&$ All For One: Bypassing Hyperparameter Tuning with Model Averaging for Cross-Lingual Transfer"
_EMNLP/2023/Conference — EMNLP 2023 Findings_

### Official Review · Reviewer_c3Ph · 2023-07-19

**Soundness:** 3

**Excitement:**

3: Ambivalent: It has merits (e.g., it reports state-of-the-art results, the idea is nice), but there are key weaknesses (e.g., it describes incremental work), and it can significantly benefit from another round of revision. However, I won't object to accepting it if my co-reviewers champion it.

**Missing References:**

Model Selection for Cross-Lingual Transfer, Chen and Ritter 2021, EMNLP (https://arxiv.org/abs/2010.06127)
Zero-shot cross-lingual transfer language selection using linguistic similarity, Eronen et al., (https://www.sciencedirect.com/science/article/pii/S030645732200351X)

**Paper Topic And Main Contributions:**

In the existing literature, the zero-shot cross-lingual transfer experiment setting often involves selecting model checkpoints based on the source language development data. However, this practice has been found to result in suboptimal performance for the target language (Keung et al., 2020; Chen and Ritter, 2021). Prior studies (Hu et al., 2020) have reported average results of model runs without any hyperparameter tuning.

To address this limitation, this paper proposes the adoption of the "model averaging" approach, wherein model checkpoints from different runs (during training) are cumulatively averaged into a single model. The experiments, which include tasks like NLI, NER, QA, demonstrate the following key findings:

1. Cumulative averaging leads to more robust cross-lingual transfer checkpoints on NLI, NER, and QA, when compared to the MAX SRC-DEV strategy.

2. The LAST/S-DEV/CA approach, coupled with cumulative averaging, yields improved performance compared to counterparts using MAX SRC-DEV.

3. Cumulative averaging with CA shows a positive correlation with the combination of Max Dev and TRG DEV ("oracle").


**Reasons To Accept:**

1. This paper proposes cumulative averaging as a means to report the performance of zero-shot cross-lingual transfer, instead of relying on the last checkpoint or selecting one based on the source language dev set.

2. The paper conducts experiments on NER, NLI, and QA, encompassing a total of (25+9+10, 25, 9) languages, with 21 configurations of hyperparameters and 3 random seeds to support the claims made in the introduction.

**Reasons To Reject:**

1. This work could benefit from the inclusion of a simple prediction ensemble baseline. Instead of averaging 10 snapshots, an alternative baseline could involve averaging the predictions from individual snapshots (for instance, employing majority votes or averaging the logits).

2. The experiments in this study focus solely on English-to-X cross-lingual transfer, disregarding standard baselines like "translate-train". Incorporating this stronger baseline as a model selection technique might result in higher performance. I am particularly intrigued to observe how this inclusion could impact the gap between S-DEV and CA.


**Reproducibility:**

4: Could mostly reproduce the results, but there may be some variation because of sample variance or minor variations in their interpretation of the protocol or method.

**Reviewer Confidence:**

4: Quite sure. I tried to check the important points carefully. It's unlikely, though conceivable, that I missed something that should affect my ratings.

**Typos Grammar Style And Presentation Improvements:**

The paper is indeed well-written; however, it could be further improved by providing a clearer explanation and distinction between "CA" (Line 186) and "Cumulative Averaging" (Lines 140-142) in Table 1. I was confused a little bit to distinguish between a weight-averaging technique and a model selection method.

---

> ### Author Rebuttal · Authors · 2023-08-28
>
> We kindly thank you for your thoughtful review. Below we provide replies to individual concerns and methodological proposals.
>
> > This work could benefit from the inclusion of a simple prediction ensemble baseline. Instead of averaging 10 snapshots, an alternative baseline could involve averaging the predictions from individual snapshots (for instance, employing majority votes or averaging the logits).
>
> This arguably is a viable alternative. The reasoning behind not analyzing logits/prediction ensembling is two-fold:
>
> First, model averaging does not increase model complexity, whereas prediction ensembling does. Crucially, prediction ensembling exacerbates computational complexity (both space and time complexity): inference requires storing "N" models and making "N" forward passes (one with each model) for the same input instance. A fair evaluation should then compare an ensemble of "N" models with a single model that is "N" times larger. LM scaling laws empirically demonstrate this works better in NLU tasks than prediction ensembles.
>
> Second, [Wang et al.](https://aclanthology.org/2022.emnlp-main.388.pdf) show that averaging adapters in loosely comparable manner beats logit-based ensembling on comparable NLU tasks.
>
> We omitted this discussion due to space constraints (after all, this is a short paper submission), but will gladly include it in the camera ready version (provided the extra page).
>
> > The experiments in this study focus solely on English-to-X cross-lingual transfer, disregarding standard baselines like "translate-train". Incorporating this stronger baseline as a model selection technique might result in higher performance. I am particularly intrigued to observe how this inclusion could impact the gap between S-DEV and CA.
>
> We agree that \`translate-train' would be quite interesting to consider. We respectfully would like to argue that the absence of \`translate-train' should not be considered a "reason to reject" our work. Cross-lingual transfer by means of translation (translate-train or translate-test) is indeed an established paradigm, one that works particularly well for transfer between high-resource languages between which MT is reliable. For low(er)-resource languages, the conventional transfer by means of multilingual LMs (the paradigm which we address in this work) is better. Besides low-resource languages, translation-based XLT is challenging for token-level tasks like NER or question-answering tasks like TyDiQA-GoldP, for which label spans must be aligned in order to do label projection.
>
> Our averaging protocol is designed for the conventional XLT transfer with multilingual LMs, which is an established and prominent XLT technique and we show that our cumulative averaging improves this established XLT approach. We would also like to point out that our submission is a short paper, the expectation of which is a focused contribution. We believe that cumulative averaging, which improves one established XLT paradigm, is exactly such a focused contribution.

---

### Official Review · Reviewer_Znmg · 2023-08-01

**Soundness:** 3

**Excitement:**

5: Transformative: This paper is likely to change its subfield or computational linguistics broadly. It should be considered for a best paper award. This paper changes the current understanding of some phenomenon, shows a widely held practice to be erroneous in someway, enables a promising direction of research for a (broad or narrow) topic, or creates an exciting new technique.

**Paper Topic And Main Contributions:**

This paper presents a comprehensive study of the evaluation protocol for cross-lingual transfer. Within this cross-lingual transfer setting where training (train), development (dev), and test data may have been written in different languages, prior studies showed that an usual procedure using the dev written in the same language with train (SRC-DEV) for model snapshot selection and hyperparameter tuning is not a good fit with respect to test time performance. Therefore, this study explored alternative strategies including the use of the last checkpoint (LAST), the dev written in the same language with test (TRG-DEV; regarded as a potential upper bound), and cumulatively averaging within-run snapshots (CA), as well as a new meta-selection proposal called accumulatively averaging.

**Questions For The Authors:**

* Q1: While I really like the paper as a whole, ~I’m pretty skeptical about the proposed CA. How is it different from other strategies like model ensembling or checkpoint averaging? IIRC those strategies can be used without looking at target-language instances. In addition, I would like to see more intuitive explanations of how CA works.~
* Q2: What’s really wrong with existing LAST and TRG-DEV? My current take is that we can use TRG-DEV if there is a target dev and LAST if there isn’t. ~CA~ accumulatively averaging seems to have a trade-off of slightly improved performance (only 0.1-ish) with the compensation of lower reproducibility over LAST due to its complexity. I would love to see more pushes for switching from LAST to ~CA~ accumulatively averaging.

**Reasons To Accept:**

* A comprehensive study of the evaluation protocol for cross-lingual transfer is presented, which is usually overlooked in practice, but it is a very important topic with respect to reproducibility.
* In particular, the literature review part stands as a good contribution to the community.

**Reasons To Reject:**

* The proposed new protocol ~CA~ accumulatively averaging seems to have some unwritten downsides due to its complexity, so the strong claim of advantages over existing protocols is a bit doubtful (see my questions).

**Reproducibility:**

5: Could easily reproduce the results.

**Reviewer Confidence:**

2: Willing to defend my evaluation, but it is fairly likely that I missed some details, didn't understand some central points, or can't be sure about the novelty of the work.

**Typos Grammar Style And Presentation Improvements:**

* C1: Standard deviation in tables is very nice information, but I have the following two doubts:
  * C1-1: Does it actually mean “standard deviation” of our model parameters? AFAIK after changing random seed used for training, outputs would no longer fall into the same standard deviation. So I personally prefer showing raw values with ranges (+-) as compensation.
  * C1-2: In addition, considering such a difficulty, is it safe to say “CA … often outperforms LAST and SRC-DEV by notable margins” (L202-204) ?

---

> ### Author Rebuttal · Authors · 2023-08-28
>
> We thank the reviewer for the comments and general appreciation of our work. We believe that there some important misunderstandings, which we attempt to clarify below. We hope that resolving this confusion will result in more favorable view of the merits of our work.
>
> > Q1: While I really like the paper as a whole, I’m pretty skeptical about the proposed CA. How is it different from other strategies like model ensembling or checkpoint averaging? IIRC those strategies can be used without looking at target-language instances. In addition, I would like to see more intuitive explanations of how CA works.
>
> We first would like to clarify that `CA` and cumulative averaging denote different concepts:
>
> - `CA` always refers to averaging checkpoints of a single training run (cf. $\S$3 Model Variants); the term is always abbreviated.
> - Cumulative averaging: average variants (i.e., `LAST`,   $...$, `CA`) of \{$1, ..., r$\} training runs with various sampled hyperparameters run-by-run without sub-selecting checkpoints on source-language validation instances; cumulative averaging is never abbreviated.
>
> We do acknowledge the unfortunate fact that cumulative averaging could also be abbreviated as CA, overloading the abbreviation. We will change cumulative averaging to "accumulative averaging" to separate the two intertwined terms more and avoid confusion.
>
> In other words, we do **not** introduce a new approach to model averaging. Our key contribution is as follows. If \{$1, ..., r$\} models are trained to maximize transfer performance (i.e., model selection over $r$ runs), it is better to cumulatively average all the $r$ `LAST`, `CA`, or `SRC-DEV` models rather than selecting the best individual `LAST`, `CA`, or `SRC-DEV` model on source-language validation instances among $r$ runs. We show that simple, (ac)cumulative averaging is crucial for model averaging in **true** zero-shot cross-lingual transfer:
>
> - Model selection on source-language validation data (\`max. src-dev') gets stuck in sub-optimal minima for both single-run variants and model averaging (i.e., see `SOUP` in Table 2)
> - (Ac)cumulative averaging closely aligns with ideal transfer performance (`TRG-DEV`)
>
> Unlike our submission, Schmidt et al. (2023) did not fairly compare single-run variants against averaging models from different runs. For a more detailed comparison, please refer to our answer of the first question of Reviewer 1.
>
> > Q2: What’s really wrong with existing LAST and TRG-DEV? My current take is that we can use TRG-DEV if there is a target dev and LAST if there isn’t. CA seems to have a trade-off of slightly improved performance (only 0.1-ish) with the compensation of lower reproducibility over LAST due to its complexity. I would love to see more pushes for switching from LAST to CA.
>
> Please note again that `CA` (i.e., checkpoint averaging over a single run) is not the contribution of this work. (Ac)cumulative averaging of checkpoints over multiple runs (for $r > 1$) is the central contribution of our work. (Ac)cumulative averaging is the strategy that can be executed over any of the single-run checkpoint selection strategies (i.e., `CA` or LAST or SRC-DEV), as visible from Table 1, where cumulative averaging has three subcolumns corresponding to the three within-run model selection strategies (LAST, S-DEV, and `CA`).
>
> That said, in the context of a single run ($r=1$), `CA` does outperform LAST by non-negligible margins. The margins are generally slimmer in higher-level tasks like NLI and generally larger in token-level tasks like NER. With cumulative averaging, `CA` frequently also frequently performs better than `LAST`.
>
> If annotated target-language instances are available (i.e., `TRG-DEV`), they should not be used for model selection but for training! The gains of model selection on such sizable target-language validation instances (often 100s of annotations) would be easily surpassed tremendously by training on target-language instances (without further validation) instead.
>
> > The proposed new protocol CA seems to have some unwritten downsides due to its complexity, so the strong claim of advantages over existing protocols is a bit doubtful (see my questions).
>
> Please see our clarification of the differences of `CA` and cumulative averaging above. We do not propose `CA` (CA was used in many works before in various tasks, from MT and in computer vision to Schmidt et al., 2023 who leverage it in the context of cross-lingual transfer). We introduce *cumulative averaging* (which we will rename to accumulative averaging to avoid confusion), a strategy that cumulatively selects and averages in models variants (individual or CA-ed checkpoints) from different runs.
>
> Our protocol is indeed somewhat more expensive than evaluating model variants ({`LAST`, ..., `CA`}) in a single-run context. However, previous work frequently performs model selection on source- or even target-language validation instances for zero-shot cross-lingual transfer over various runs. The core argument of our approach is that, if we are already retraining the model several times for model selection, we achieve better (or even ideal `TRG-DEV`-like) transfer performance without target-language instances by cumulatively averaging the resulting runs, rather than identifying the best model within those runs.

---

### Official Review · Reviewer_H7uL · 2023-08-07

**Soundness:** 4

**Excitement:**

3: Ambivalent: It has merits (e.g., it reports state-of-the-art results, the idea is nice), but there are key weaknesses (e.g., it describes incremental work), and it can significantly benefit from another round of revision. However, I won't object to accepting it if my co-reviewers champion it.

**Paper Topic And Main Contributions:**

Increasing score after rebuttals ... assuming the authors will add the clarifications into the paper

``````````````````````````````````````````````````````


This paper proposes a solution to the problem of over-fitting on source languages in zero-shot cross-lingual transfer by averaging models. In general, this is one of these papers whose benefits lie in its simplicity. By simply showing that model averaging, common in other NLP tasks, such as MT which they identify, can outperform more complicated methods that rely on more data and are tough to reproduce, the paper is a useful contribution to the field. The authors do a good job motivating the paper with why zero-shot doesn’t work – including highlighting issues with using validation examples in the target languages. However, my biggest concern with the paper is baseline comparisons and situating the work in relation to other work – particularly Schmidt et al. (2023).

In Schmidt et al. (2023), which the authors cite extensively, there is Checkpoint Averaging (CA) and Run Averaging  (RA). As an initial point, the CA acronym appears to be a bit overloaded and that this work, which also uses CA, appears to be a bit closer to their RA. It would help a lot to clear this up in the writing, or to use a different acronym.

Secondly, since the authors cite Schmidt et al. (2023) so extensively, and mention that it motivates some of their research questions, I would have liked to have seen a more clear comparison their work and to use their work as a baseline. Normally, I would not ask a submission to reproduce work that has been on arxiv for only a few weeks before this conference’s deadline, but the fact that the paper is cited 10 times in this submission, makes me want to see an empirical comparison.

The bulk of my review basically boils down to a few questions that I would like to see answered by the authors in the rebuttal period:
•	How exactly is your sampling method different from Schmidt et al. (2023)? Particularly the RA and RA-CA variants of that paper?
•	What are the differences in scores between the methods in Schmidt et al. (2023) and your method? I can see tables in both for similar tasks, but very different numbers. How am I supposed to compare them? Ideally, running experiments with the same setup (aka pre-processing, tokenization, etc.) would be better than just reporting numbers from another paper. Since the methods are so similar, this should not be a large ask. This should be shown in a table in the main part of the paper (not an appendix).
•	What is the difference in runtimes of the methods? It is unclear to me how computationally expensive this method is compared to Schmidt et al. (2023). In particular, their definition of CA seems like it is only one run, so it would be much less computationally complex.

Overall, I liked the paper and would be happy to increase my score after author rebuttal if the above questions are answered. The paper has a lot of potential – but until the differences with a paper cited 10 times are reconciled, it is hard to give a good score.

**Questions For The Authors:**

•	How exactly is your sampling method different from Schmidt et al. (2023)? Particularly the RA and RA-CA variants of that paper?
•	What are the differences in scores between the methods in Scmidt et al. (2023) and your method? I can see tables in both for similar tasks, but very different numbers. How am I supposed to compare them? Ideally, running experiments with the same setup (aka pre-processing, tokenization, etc.) would be better than just reporting numbers from another paper. Since the methods are so similar, this should not be a large ask. This should be shown in a table in the main part of the paper (not an appendix).
•	What is the difference in runtimes of the methods? It is unclear to me how computationally expensive this method is compared to Schmidt et al. (2023). In particular, their definition of CA seems like it is only one run, so it would be much less computationally complex.

**Reasons To Accept:**

This is a useful contribution to the field that demonstrates a simpler, and more effective way, of doing zero-shot cross-lingual transfer.

**Reasons To Reject:**

The model is not compared with (or is unclear if it is) with the baselines of Schmidt et al. (2023)

**Reproducibility:**

4: Could mostly reproduce the results, but there may be some variation because of sample variance or minor variations in their interpretation of the protocol or method.

**Reviewer Confidence:**

5: Positive that my evaluation is correct. I read the paper very carefully and I am very familiar with related work.

**Typos Grammar Style And Presentation Improvements:**

Section 2 could define variables a bit more clearly.

Section 3. Training Details. “We train XLM-R….” I think this should be “We fine-tune XLM-R…”

---

> ### Author Rebuttal · Authors · 2023-08-28
>
> We thank you for your comprehensive review and for asking specific clarification questions! We believe that our replies will clarify most of the points and lead to more favorable recommendation scores.
>
> > As an initial point, the CA acronym appears to be a bit overloaded and that this work, which also uses CA, appears to be a bit closer to their RA. It would help a lot to clear this up in the writing, or to use a different acronym.
>
> We would first like to clarify this issue. Please note that CA and cumulative averaging are distinct terms.
>
> - `CA` (checkpoint averaging): a model "variant" which averages checkpoints of a single training run (cf. $\S$3 Model Variants); the term is always abbreviated.
> - Cumulative averaging: average variants (i.e., `LAST`,
> $...$, `CA`) of {$1, ..., r$} training runs with various sampled hyperparameters run-by-run without sub-selecting checkpoints on source-language validation instances (cf. $\S$2 Cumulative Run Averaging); cumulative averaging is never abbreviated.
>
> We do acknowledge the unfortunate fact that cumulative averaging could also be abbreviated as CA, overloading the abbreviation. We will change cumulative averaging to "accumulative averaging" to separate the two intertwined terms more and avoid confusion.
>
> > How exactly is your sampling method different from Schmidt et al. (2023)? Particularly the RA and RA-CA variants of that paper? What are the differences in scores between the methods in Schmidt et al. (2023) and your method?
>
> Schmidt et al. (2023) report performance of `LAST`, `SRC-DEV`, and `CA` on a *single hyperparameter configuration*, averaged over 5 random seeds (these are the different runs, which they average). This benefits `RA` (i.e., run averaging over 5 runs) over `LAST` (last checkpoint), `SRC-DEV` (optimal checkpoint according to source language dev set performance), and `CA` (average of all checkpoints of the run). There is no model/checkpoint selection over all 5 runs together.
>
> In contrast, we more realistically benchmark the single-run \{`LAST`, ..., `CA`\} model variants of $1, ..., r$ runs with *different hyperparameter configurations* that perform best on source-language validation data -- i.e., selecting the best possible checkpoint from all checkpoints of all $r$ runs (this is our primary baseline) -- against our simple, (ac)cumulative averaging in true zero-shot cross-lingual transfer.
>
> The results show that simple, (ac)cumulative averaging is a crucial for model averaging in zero-shot cross-lingual transfer:
>
> - Model selection on source-language validation data (\`max. src-dev') gets stuck in sub-optimal minima for both single-run variants and model averaging (`SOUP`)
> - (Ac)cumulative averaging closely aligns with ideal transfer performance (`TRG-DEV`)
>
> As the reviewer correctly asserts, we closely mimic the experimental setup of Schmidt et al., 23 (i.e., model variants). The difference in scores stems from running experiments with a different multilingual Transformer as a backbone: we use XLM-R-Large rather than XLM-R-Base (Schmidt et al.) given the recent prevalence of larger LMs.
>
> > I can see tables in both for similar tasks, but very different numbers. How am I supposed to compare them?
>
>  **To preface: where applicable, our results are in line with Schmidt et al. (2023).**
>
> As model sizes differ (theirs: XLM-R-Base; ours: XLM-R-Large), we can only (roughly) compare the relative margins in performance task-by-task between different model variants (i.e., {`LAST`, $...$, `CA`}) between their results and our results.
>
> One **loose comparison** would be to compare $r=1$ of Table 1 in our submission to Table 1 of Schmidt et al. (2023). $r=1$ reflects the average performance of 10 single runs with varying hyperparameters. Table 1 of Schmidt et al. (2023) reports averages of 5 runs with a single set of hyperparameters. `RA` or `SOUP` of Schmidt et al. could (loosely) be compared against our (ac)cumulative averaging at $r=5$.
>
> > Ideally, running experiments with the same setup (aka pre-processing, tokenization, etc.) would be better than just reporting numbers from another paper. Since the methods are so similar, this should not be a large ask.
>
> Barring evaluating on XLM-R-large rather than XLM-R-base, our experimental setup is aligned with Schmidt et al. (2023). The key difference is how performance of models by averaging checkpoints stored in various runs (theirs: `RA`, ours: (ac)cumulative averaging) is presented. As mentioned above, they re-run training on the same set of hyperparameters. We simulate re-training models with highly diverse learning rates and batch sizes and cumulatively average \{`LAST`, $...$, `CA`\} of these runs.
>
> > What is the difference in runtimes of the methods? It is unclear to me how computationally expensive this method is compared to Schmidt et al. (2023). In particular, their definition of CA seems like it is only one run, so it would be much less computationally complex.
>
> Crucially, cumulative averaging does not incur additional compute costs to standard hyperparameter tuning. The only cost comes from re-training models, which, if hyperparameters are tuned, is an expense that is incurred anyways. With respect to a single run (cf. $r=1$), we confirm that `CA`, in expectation, outperforms `LAST` and `SRC-DEV` in **true** zero-shot cross-lingual transfer. While (ac)cumulative averaging over multiple runs is clearly more computationally demanding than CA over a single run (r = 1 in Table 1) it brings substantial performance gains already for r = 2.

---

### Meta-Review · Area_Chair_bUFx · 2023-09-19

**Recommendation:** 3

**Metareview:**

This paper proposes a cumulative averaging of model snapshots from different runts. This aims to avoid the need for extensive hyper-parameter search for zero-shot tasks. Overall the idea is innovative and might open the doors for future research. However, the reviewers also found a number of issues: lack of proper testing on standard methods like ``English-to-X cross-lingual transfer, disregarding standard baselines like "translate-train"''(Rev c3Ph); an unlcear writing. In general, having a stronger comparison with baselines could help to improve the paper. That said, the authors  clarified some not clear aspects of the paper.

---

### Decision · Program_Chairs · 2023-10-07

**Decision:**

Accept-Findings

**Comment:**

This paper proposes a cumulative averaging of model snapshots from different runts. This aims to avoid the need for extensive hyper-parameter search for zero-shot tasks. Overall the idea is innovative and might open the doors for future research. However, the reviewers also found a number of issues: lack of proper testing on standard methods like ``English-to-X cross-lingual transfer, disregarding standard baselines like "translate-train"''(Rev c3Ph); an unlcear writing. In general, having a stronger comparison with baselines could help to improve the paper. That said, the authors  clarified some not clear aspects of the paper.